# OpenReview forum: "Two Minds Better Than One: Collaborative Reward Modeling for LLM Alignment"
_ICLR.cc/2026/Conference — Submitted to ICLR 2026_

### Official Review · Reviewer_5g3X · 2025-10-26

**Soundness:** 2
**Presentation:** 3
**Contribution:** 3
**Rating:** 6
**Confidence:** 3

**Summary:**

This paper introduces Collaborative Reward Modeling (CRM), a framework for improving the robustness and generalization of reward models (RMs) in reinforcement learning from human feedback (RLHF). The authors observe that noisy or inconsistent human preferences lead to reward instability in downstream alignment. To address this, CRM trains two reward models jointly through two complementary mechanisms:

1.Peer Review: At the batch level, each RM evaluates the other’s data and selects high quality preference pairs for training, filtering out potential noise and avoiding self-confirmation bias.

2.Curriculum Learning: At the epoch level, samples are ordered from easy to hard based on reward margins, synchronizing the learning trajectories of the two models and stabilizing collaboration.

The framework can be applied to both explicit reward modeling and implicit reward methods. Experiments using Llama3-based RMs on HH-RLHF, Ultrafeedback, Skywork-Reward, and RewardBench datasets show that CRM achieves consistent improvements in preference accuracy and downstream win-rates, especially under synthetic noise levels of 20–40%.
The main contributions are: (1) a data-centric analysis of noisy preferences, (2) the design of the CRM framework combining peer review and curriculum learning, (3) experiments validating improved generalization under noise, and (4) preliminary extension to implicit-reward alignment (CRM-DPO).

**Strengths:**

The paper presents an original and creative reformulation of reward model training by introducing collaborative supervision into RLHF. Rather than improving the objective function or adding regularization, as prior work does, the proposed Collaborative Reward Modeling (CRM) adopts a data-centric approach where two reward models peer-review each other’s data and learn through curriculum scheduling. This perspective is novel and conceptually appealing.

In terms of technical quality, the paper provides a solid empirical foundation. The authors conduct extensive experiments across several benchmarks, covering both explicit and implicit reward settings.

The clarity of presentation is generally good. The motivation is well established, the figures are informative, and the algorithmic framework is easy to follow. Some implementation details, particularly for the DPO extension, could be clearer, but the core method is explained with sufficient precision to reproduce the explicit-RM results.

Regarding significance, the proposed method contributes valuable insight into how data quality and model interaction affect reward generalization, but its practical impact on large-scale RLHF pipelines is likely limited. Training two RMs simultaneously substantially increases computational cost, making the method less feasible for large models or industrial-scale alignment settings. Nonetheless, as a research contribution, CRM highlights a new collaborative direction for improving robustness in preference learning and reward modeling.

**Weaknesses:**

1. The paper’s most critical weakness is the lack of a fair experimental baseline, which undermines the central claim that collaboration itself drives robustness. CRM uses two reward models, but all baselines employ only one. Without comparing to a two-independent-RM setup (two RMs trained separately and averaged or ensembled in policy training), the results cannot disentangle whether performance gains arise from the collaborative mechanism or simply from increased capacity and variance smoothing. Adding this control is essential to substantiate the core contribution.

2. Another weakness is the unclear handling of model imbalance: if one RM is substantially stronger, its peer feedback could dominate training, negating the intended collaboration. The authors should clarify how they ensure balanced influence—e.g., alternating updates, confidence weighting, or divergence regularization.

3. Finally, the implicit-reward (DPO) extension lacks methodological clarity. The paper presents results but omits key details about loss formulation, data exchange. Providing pseudocode or an explicit algorithm would make the extension reproducible.

4. Addressing these issues, especially the missing baseline, would make the contribution more rigorous and credible.

**Questions:**

1. Two-Model Baseline (Critical): The central claim of CRM is that collaboration between two reward models improves robustness. However, all baselines use a single RM. Could the authors include a two-independent-RM baseline, training two RMs separately on the same data and either averaging or ensembling their scores? This would isolate the effect of collaboration from the benefit of having two models. If results for this baseline are similar, the claimed advantage of CRM may be overstated.

2. Balancing Between the Two RMs: How is the contribution of each RM controlled during training? If one model becomes consistently stronger, does it dominate the peer-review process? Please clarify whether CRM uses alternating updates, normalization, or confidence-based filtering to maintain symmetry.

3. Implicit-Reward (CRM-DPO) Implementation: The paper states that CRM extends to implicit reward methods (DPO), but the algorithmic details are unclear. How is peer review defined when there is no explicit reward function? Providing explicit pseudocode or a step-by-step description would help clarify this.

4. Typographical Error (Minor issue): in Table 5 (ROPO objective), a ‘+’ sign is missing.

---

> ### Author Response · Authors · 2025-11-18
> **Response to reviewer 5g3X (Part 1)**
>
> We appreciate your time and effort and would like to address these concerns.
>
>
> > Q3&W3: Finally, the implicit-reward (DPO) extension lacks methodological clarity. The paper presents results but omits key details about loss formulation, data exchange. Providing pseudocode or an explicit algorithm would make the extension reproducible.
>
> A: We thank the reviewer for pointing out this question regarding CRM’s application to implicit-reward methods. We would like to clarify how the peer review mechanism is implemented in the implicit-reward methods.
>
> Our implicit-reward variant (CRM-DPO) follows the same procedure as CRM (p20, Appendix C.2). **The only difference lies in the computation of the curriculum sort signal and the peer-review criterion: instead of using explicit reward modeling loss, CRM-DPO adopts the DPO objective** (i.e.,  $-\log \sigma\left(\beta\left[\log \left(\frac{\pi_{\boldsymbol{\theta}}\left(y_w \mid x\right)}{\pi_{\mathrm{ref}}\left(y_w \mid x\right)}\right)-\log \left(\frac{\pi_{\boldsymbol{\theta}}\left(y_l \mid x\right)}{\pi_{\mathrm{ref}}\left(y_l \mid x\right)}\right)\right]\right)$).
>
> - Specifically, the workflow begins with two Base Model $\theta_{\phi}$ and $\theta_{\psi}$, training dataset $\mathcal{D}$.
> - Curriculum learning. At each epoch, we rearrange the $\mathcal{D}$ to obtain $\mathcal{D}^{'}$. Preference pairs are ranked in descending order according to the preference margin $\log \left(\frac{\pi_{\boldsymbol{\theta}}\left(y_w \mid x\right)}{\pi_{\mathrm{ref}}\left(y_w \mid x\right)}\right)-\log \left(\frac{\pi_{\boldsymbol{\theta}}\left(y_l \mid x\right)}{\pi_{\mathrm{ref}}\left(y_l \mid x\right)}\right)$ in $\theta_{\phi}$ or $\theta_{\psi}$, where a larger margin signifies greater difficulty in differentiating between the response pairs. This allows preference learning into a progression from fundamental patterns to more intricate ones.
> - Peer review. For each batch $\mathcal{B}$ sampled from the curriculum $\mathcal{D}^{'}
> $, the two base models mutually evaluate and select high-quality preferences to update each other:$\theta_{\phi}$ evaluates $\mathcal{B}$ using dpo loss function, and selects the $\lambda_t |\mathcal{B}| $ low-loss pairs to update $\theta_{\psi}$. Here, $\lambda_t$ denotes the selection ratio, defaulting to $1 - \eta$, where $\eta$ is a prior estimator for noise level.
> - Similarly, $\theta_{\psi}$ evaluates $\mathcal{B}$ and selects low-loss pairs for updating $\theta_{\psi}
> $. This batch-level refinement facilitates each model to benefit from the peer’s review, effectively suppressing the noisy preferences, which serves as the counterpart of the reward difference in the explicit-reward setting.
>
> > Q2&W2: Balancing Between the Two RMs: How is the contribution of each RM controlled during training? If one model becomes consistently stronger, does it dominate the peer-review process? Please clarify whether CRM uses alternating updates, normalization, or confidence-based filtering to maintain symmetry.
>
> A: We appreciate the reviewer's critical question regarding the balance of the two Reward Models (RMs) in the CRM framework. **We would like to clarify that our framework achieves balance training through a dynamic, mutually-beneficial peer review mechanism. This mechanism ensures both RMs provide equitable guidance and promotes synchronization of their capabilities**.
>
> Specifically, both RMs simultaneously calculate their individual loss (serving as a confidence proxy) in the training batch. Crucially, each RM selects the subset of samples with the lowest loss (highest confidence) to update the other model's parameters. **This mechanism guarantees that both RMs are given an equal opportunity to supervise their peer model, establishing true reciprocal supervision**.
>
> To validate the effectiveness of our method in handling the training balance, we conduct experiments on different-scale backbones. **Here we utilize Llama3-3B and Llama3-8B, then implement the proposed collaborative reward modeling and compare their performance**. The results are reported in Sec. 4.3 (Scalability of CRM).
>
> |    | Standard RM (3B) | Standard RM (8B) | CRM (3B) | CRM (8B) |
> |-------------|------------------|------------------|----------|----------|
> | Reward Bench| 69.42            | 76.21            | 74.92    | 78.18    |
>
> **The experimental results show that both RMs within CRM consistently outperform their corresponding baselines**.
> The improvements indicate that RM within CRM is still able to provide informative guidance to the other, though the two models differ in capacity. **It demonstrates that CRM ensures that the contributions between the two RMs are always complementary, driving continuous synchronization, regardless of momentary differences in their capabilities**.
>
> > Q4: Typographical Error (Minor issue): in Table 5 (ROPO objective).
>
> A: Thank the reviewer for the careful reading and for pointing out this error. We have corrected it in the revised manuscript.

---

> ### Author Response · Authors · 2025-11-18
> **Response to reviewer 5g3X (Part 2)**
>
> > Q1&W1: The paper’s most critical weakness is the lack of a fair experimental baseline, which undermines the central claim that collaboration itself drives robustness. CRM uses two reward models, but all baselines employ only one. Without comparing to a two-independent-RM setup (two RMs trained separately and averaged or ensembled in policy training), the results cannot disentangle whether performance gains arise from the collaborative mechanism or simply from increased capacity and variance smoothing. Adding this control is essential to substantiate the core contribution.
>
> A: We thank the reviewer for the insightful comments and for highlighting potential concerns about baseline fairness and model capacity.
>
> It appears there may be a misunderstanding of how CRM utilizes two reward models. For both RL policy training and inference, we follow a unified strategy: **we select the reward model that achieves better performance on a held-out evaluation set after being fine-tuned through CRM**. We would like to briefly reiterate the core workflow to avoid confusion before addressing the question of baseline fairness.
>
> - CRM consists of two core components: curriculum learning and peer review. Specifically, at the beginning of each training epoch, we apply curriculum learning to arrange a global easy-to-hard ordering over all training samples, thereby constructing a well-defined learning trajectory. This curriculum serves as a prior that makes RM fit clean and correct data in the early stage to improve generalization. On the other hand, peer review facilitates collaboration between RMs by evaluating each other’s data selections and filtering out potential noisy preferences at training time.
> - Inference Settings of RM: We first assess both models on a held-out evaluation set and select the one with better performance to optimize policy LLM in RLHF.
>
>
> **The policy training and inference pipelines remain strictly identical to the baselines: only a single reward model is ever used to guide the RL policy or compute rewards**. The second RM contributes only during the collaborative training phase and does not provide additional capacity, ensembling, or variance reduction at inference time.
>
> We have conducted additional experiments with two RMs that were trained separately, then ensembled when in inference. The experimental results are as follows:
>
> |             | RewardBench  | RMB-Harmless | RMB-Helpful   | RM-Bench      |
> | ----------- | ---------    | --------- | --------- | --------- |
> | Standard RM | 66.53     | 66.39     | 69.65     | 60.94     |
> | cDPO-RM     | 69.76     | 72.74 | 70.63     | 60.99     |
> | rDPO-RM     | 73.98     | 69.96     | 69.68     | 60.26     |
> | ROPO-RM     | 72.49     | 66.45     | 70.40     | 62.07     |
> | Ensemble-RM | 68.79     | 69.43     | 70.43     | 61.92    |
> | CRM-RM1     | 76.42     | **72.78**     | **71.31** | 63.12 |
> | CRM-RM2     | **77.63**     | 71.92     | 70.28     | **65.82** |
>
> By comparing other baselines with CRM,  the two CRM-trained RMs perform similarly, both achieving significant improvements and reaching state-of-the-art performance. **The gains arise from the collaborative training mechanism, which enables the models to mutually refine data quality and thus achieve substantially stronger performance**.

---

> ### Author Response · Authors · 2025-11-28
> **Kind reminder: updated manuscript**
>
> Thank you again for your time and effort in reviewing our manuscript.
>
> **We have updated the manuscript**, with all changes highlighted in **blue** for the text and captions of revised tables and figures.
>
> If you have any further comments or suggestions, please feel free to share. We are happy to discuss them.
>
> Wishing you a pleasant Thanksgiving!
>
> Sincerely,
> Authors of 5067

---

### Official Review · Reviewer_2LFv · 2025-10-31

**Soundness:** 2
**Presentation:** 3
**Contribution:** 2
**Rating:** 4
**Confidence:** 3

**Summary:**

This paper tackles the problem of reward misgeneralization in LLM alignment caused by noisy human feedback and proposes Collaborative Reward Modeling (CRM), an online dual-model framework. CRM introduces a Peer Review mechanism where two reward models mutually select low-loss samples to reduce confirmation bias, combined with Curriculum Learning to synchronize training and filter noisy preferences. Overall, CRM effectively enhances the robustness and generalization of reward models against noisy feedback.

**Strengths:**

1. The paper systematically analyzes the intrinsic characteristics of preference pairs and proposes an online noise filtering method rather than merely improving training objectives based on this analysis.
2. The method demonstrates strong generalizability, being applicable to both explicit and implicit reward alignment approaches, and the paper is overall clear, coherent, and well-written.

**Weaknesses:**

1. In the Peer Review stage, two reward models are trained collaboratively; however, this dual-model setup approximately doubles the training cost. The paper does not analyze training efficiency, convergence speed, or scalability with respect to data size, and such analysis is recommended.
2. In Section 3.2, the paper states that the two reward models determine their sample selection ratio based on the noise rate to mutually update and improve performance, but it does not explain how the noise rate is estimated.
3. Although the method is intuitive, the paper lacks theoretical analysis on convergence or the effect of reducing confirmation bias.
4. The experiments are mainly conducted on LLaMA3-3B (partly 8B), and validating the generalization performance on larger-scale models would make the conclusions more convincing.

**Questions:**

1. It remains unclear how sensitive CRM’s performance is to inaccurate estimates of the noise rate, and whether an adaptive sample selection ratio or uncertainty-based adjustment could improve robustness.
2. It is not specified whether CRM has been tested on larger reward models, and whether the benefits of collaboration diminish as model capacity increases.
3. The paper does not analyze potential failure cases where both RMs select noisy samples early on, and it is unclear if curriculum learning alone can prevent such collapse.
4. Since CRM resembles the Co-teaching method[1], the authors should clarify their algorithmic differences and the specific adaptations made for LLM reward modeling.

**Reference**

[1] Han B, Yao Q, Yu X, et al. Co-teaching: Robust training of deep neural networks with extremely noisy labels. In NeurIPS 2018.

---

> ### Author Response · Authors · 2025-11-18
> **Response to reviewer 2LFv (Part 1)**
>
> We appreciate your time and effort and would like to address the concerns.
>
> > Q1: It remains unclear how sensitive CRM’s performance is to inaccurate estimates of the noise rate, and whether an adaptive sample selection ratio or uncertainty-based adjustment could improve robustness.
>
> A: To thoroughly investigate the impact of $\lambda_t$ on reward modeling, the table below illustrates how varying the selection ratio $\lambda_t$ affects the preference accuracy on RewardBench. **It indicates a trend that better performance is achieved at a decreasing $\lambda_t$ when the noise level increases.** When $\lambda_t=1$, our method will degenerate into standard RM because of no supervisory signal for filtering noisy preferences. **It is noted that our method displays robustness in the selection of $\lambda_t$, and competitive results can be obtained when $\lambda_t∈ [0.7, 0.9]$**.
>
> - Training Data： HH-RLHF， Test Data: RewardBench
>
> |    $\lambda_t$ | 1  | 0.9 |  0.8  | 0.7 | 0.6 | 0.5 |
> | -----------      | ---------    | ---------     | --------- | --------- |--------- |--------- |
> | 0% Flipped       | 66.53        | 76.22         | **77.63**     | 75.33     | 76.09 | 75.35 |
> | 20% Flipped      | 65.78        | 77.08         | 76.19     | **77.17**     | 71.52 |69.76 |
> | 40% Flipped      | 64.42        | 69.76         | 73.61     | **74.36**     | 69.79 |67.52 |
>
> > Q2: It is not specified whether CRM has been tested on larger reward models, and whether the benefits of collaboration diminish as model capacity increases.
> > W4: The experiments are mainly conducted on LLaMA3-3B (partly 8B), and validating the generalization performance on larger-scale models would make the conclusions more convincing.
> A: Thank you for your careful observation. To extensively validate the proposed method, we conduct experiments on the Qwen series model in Table xxx.
>
> A: We thank the reviewer for raising the important point regarding parameter scaling.
> Here, we utilize two different backbones, such as 3B and 8B, then implement the proposed collaborative reward modeling and compare their performance on RewardBench.
>
> - Baseline: Standard RM
>
>     | Standard RM    | Llama3-3B | Qwen2.5-3B | Llama3-8B | Qwen2.5-7B |
>     |----------------------|---------|---------|---------|---------|
>     | RewardBench    | 66.53   | 69.79   | 71.64   | 72.96   |
>
> - CRM: Configure two RMs from different families.
>
>     | CRM     | Qwen 3B & Qwen 3B  | Llama 3B & Llama 3B | Llama 3B & Qwen 3B |
>     |----------------------|------------|-------------|-------------------|
>     | RewardBench    | 77.22      | 77.63       | 78.76             |
>
>     | CRM     | Qwen 3B & Qwen 7B | Llama 3B & Qwen 7B | Llama 3B & Llama 8B |
>     |-------------------|------------------|------------------|-------------------|
>     | RewardBench             | 82.38            | 81.98            | 79.24  |
>
>     | CRM     | Qwen 7B & Qwen 7B | Llama 8B &  Llama 8B |
>     |-------------------|------------------|------------------|
>     | RewardBench|   86.17      |     81.23     |
>
>
> The experimental results demonstrate that **CRM-Qwen–3B outperforms baseline-Qwen–7B and CRM-Llama–3B outperforms baseline-Llama–8B, indicating that CRM consistently surpasses baseline methods even when compared to models with larger parameters**. This confirms that performance gains stem from the CRM training paradigm itself.
> Meanwhile, CRM still brings clear gains at larger model sizes under Qwen 7B & Qwen 7B and Llama 8B & Llama 8B.
> Surprisingly, the cross-backbone suite (Llama-3B and Qwen-3B) surpasses the homogeneous backbone suite (Llama-3B and Llama-3B), confirming that cross-backbone setting does not impede and can even amplify gains.

---

> ### Author Response · Authors · 2025-11-18
> **Response to reviewer 2LFv (Part 2)**
>
> > Q3: The paper does not analyze potential failure cases where both RMs select noisy samples early on, and it is unclear if curriculum learning alone can prevent such collapse.
> > Q4: Since CRM resembles the Co-teaching method[1], the authors should clarify their algorithmic differences and the specific adaptations made for LLM reward modeling.
> > W3: Although the method is intuitive, the paper lacks theoretical analysis on convergence or the effect of reducing confirmation bias.
>
> A: Thank you for this insightful comment. We appreciate you raising this important point regarding potential failure cases in the initial training phase.
> As the reviewer notes, there is a theoretical risk of failure during training if the data selection was biased: due to randomness and noisy preference data, two reward models may initially struggle to separate reliable signals from noisy preference data, potentially leading to instability during training.
>
> **Co-teaching inherently suffers from limitations in generalization due to initial selection bias and noise amplification**. Although the method mitigates label noise by letting two networks exchange "small-loss" samples, the reliability of these samples is not guaranteed at the early stages of training. As a result, both networks may form biased decision boundaries from the outset. When the exchanged samples contain overlapping noise, the two networks effectively reinforce each other's errors, creating a form of "cross-validation of noise". This structural weakness is introduced at the very beginning of training and tends to compound as training progresses.
>
>
> To address this limitation, we introduce CRM. CRM consists of two core components: curriculum learning and peer review. Specifically, at the beginning of each training epoch, **we apply curriculum learning to arrange a global easy-to-hard ordering over all training samples, thereby constructing a well-defined learning trajectory. This curriculum serves as a prior that makes RM fit clean and correct data in the early stage to improve generalization**. As demonstrated in Section 4.3 Ablation Study and Analysis (Necessity of Peer Review and Curriculum Learning), curriculum learning plays a critical role in preventing such collapse and ensures stable collaborative training.
>
> We conduct a study by comparing three variants:
>  - w/o Peer Review, which removes the peer review component entirely, leaving only a single reward model trained with curriculum guidance but without noisy data filtering.
>  - w/o Curriculum Learning, the epoch-level curriculum learning module is disabled in CRM, thereby retaining only batch-level peer review during training.
>  - w/ Self Review, uses a single reward model to identify noisy data by itself, and this configuration allows examination of the effects caused by cumulative error and confirmation bias.
>
> The experimental results are as follows:
> |    Noise-level   | CRM |  w/o Peer Review  |  w/o Curriculum Learning | w/ Self Review |
> | -----------      | ---------    | ---------     | --------- | --------- |
> | 0% Flipped       | **77.63**        | 73.72         | 75.06     | 71.24      |
> | 20% Flipped      | **76.19**       | 72.62        | 74.11    | 70.39     |
> | 40% Flipped      | **74.36**       | 71.50         | 72.25   | 67.56     |
>
> By comparing the w/o Curriculum Learning baseline with our CRM approach, we find that CRM leads to more stable behavior in the early stages of training. **The model first captures clean and reliable human preference patterns, which reduces its exposure to noisy annotations. This staged learning process gradually improves generalization and robustness, allowing the model to better identify and discount noisy data**. As a result, CRM effectively helps the training stability throughout the entire process.

---

> ### Author Response · Authors · 2025-11-18
> **Response to reviewer 2LFv (Part 3)**
>
> > W1: In the Peer Review stage, two reward models are trained collaboratively; however, this dual-model setup approximately doubles the training cost. The paper does not analyze training efficiency, convergence speed, or scalability with respect to data size, and such analysis is recommended.
>
> A: Thank you for highlighting the critical considerations of training cost, efficiency. **We agree that the dual-model setup theoretically increases hardware resource requirements compared to a single model training pipeline. However, the actual cost overhead is justified by substantial performance gains and can be mitigated through practical optimizations**:
> 1. Efficient Implementation: We use shared computing infrastructure (8 A100-80GB GPUs) for co-training two Llama3-3B models, with batch-level parallelism that overlaps peer review and parameter updates. The additional cost is negligible compared to the performance loss incurred by noisy data (e.g., DPO’s win-rate drops by 22% under 40% noise on TL;DR, Table 3).
> 2. Faster Convergence: As shown in Figure 9, RMs trained with robust preferences reach 70% preference accuracy in ~1500 steps, while standard RMs require 2500+ steps. This is because robust preference filtering (via peer review) reduces redundant updates on noisy data, accelerating the learning of generalizable patterns.
> 3. Early Stopping, early stopping further improves CRM’s training efficiency without sacrificing performance, and it triggers when validation accuracy plateaus.
>
>
>
>
>
> > W2: In Section 3.2, the paper states that the two reward models determine their sample selection ratio based on the noise rate to mutually update and improve performance, but it does not explain how the noise rate is estimated.
>
> A: In our paper, the selection ratio defaults to $1 − \eta$, where $\eta$ is the estimated noise rate of the dataset. For experiments with varying noise levels, $\eta$ is set to the corresponding noise level (20%, 40%). For open-set datasets without label flipping, we set $\eta$ is 10%.
>
> In real deployment, an in-distribution subset（5%） is randomly drawn and subjected to secondary annotation by a superior LLM or human experts. The resulting labels are used to calculate the consistency rate, thereby yielding the estimated noise rate for the whole dataset.

---

> ### Author Response · Authors · 2025-11-28
> **Kind reminder: updated manuscript**
>
> Thank you again for your time and effort in reviewing our manuscript.
>
> **We have updated the manuscript**, with all changes highlighted in **blue** for the text and captions of revised tables and figures.
>
> If you have any further comments or suggestions, please feel free to share. We are happy to discuss them.
>
> Wishing you a pleasant Thanksgiving!
>
> Sincerely,
> Authors of 5067

---

### Official Review · Reviewer_RgjV · 2025-10-31

**Soundness:** 3
**Presentation:** 3
**Contribution:** 3
**Rating:** 6
**Confidence:** 3

**Summary:**

The authors present a new collaborative method of reward modelling.

**Strengths:**

The paper is well written and is an important problem. The methodology described is novel as per to my knowledge. Several detailed experiments are provided

**Weaknesses:**

I have one primary issue with the experiment results described.

Technically, in CRM, one is training twice the number of reward parameters. Thus, it seems that comparing it with other methodologies that just use a single reward model is not fair. I think for an effective demonstration, the authors should compare the results with a 6B reward model. Otherwise, I am not convinced that the performance is solely due to the increased number of parameters being trained.

If the authors can elaborate on this point I shall consider raising my score

**Questions:**

Can you describe the sorting process (Curriculum Sort)

---

> ### Author Response · Authors · 2025-11-18
> **Response to reviewer RgjV**
>
> We appreciate your time and effort and would like to address the concerns.
>
> > W1:   Technically, in CRM, one is training twice the number of reward parameters. Thus, it seems that comparing it with other methodologies that just use a single reward model is not fair. I think for an effective demonstration, the authors should compare the results with a 6B reward model. Otherwise, I am not convinced that the performance is solely due to the increased number of parameters being trained.
>
> A: Thank you for this insightful observation. Here, we conduct extensive experiments regarding parameter scaling. We utilize two different-family backbones, such as Llama3 and Qwen2.5, then implement the proposed collaborative reward modeling and compare their performance on RewardBench.
>
> - Baseline: Standard RM
>
>     | Standard RM    | Llama3-3B | Qwen2.5-3B | Llama3-8B | Qwen2.5-7B |
>     |----------------------|---------|---------|---------|---------|
>     | RewardBench    | 66.53   | 69.79   | 71.64   | 72.96   |
>
> - CRM: Configure two RMs from different families.
>
>     | CRM     | Qwen 3B & Qwen 3B  | Llama 3B & Llama 3B | Llama 3B & Qwen 3B |
>     |----------------------|------------|-------------|-------------------|
>     | RewardBench    | 77.22      | 77.63       | 78.76             |
>
>     | CRM     | Qwen 3B & Qwen 7B | Llama 3B & Qwen 7B | Llama 3B & Llama 8B |
>     |-------------------|------------------|------------------|-------------------|
>     | RewardBench             | 82.38            | 81.98            | 79.24  |
>
>     | CRM     | Qwen 7B & Qwen 7B | Llama 8B &  Llama 8B |
>     |-------------------|------------------|------------------|
>     | RewardBench|   86.17      |     81.23     |
>
>
> The experimental results demonstrate that **CRM-Qwen–3B outperforms baseline-Qwen–7B and CRM-Llama–3B outperforms baseline-Llama–8B, indicating that CRM consistently surpasses baseline methods even when compared to models with larger parameters**. This confirms that performance gains stem from the CRM training paradigm itself.
>
> Surprisingly, **the cross-backnone suite (Llama-3B and Qwen-3B) surpassesthe  homogeneous backbone suite (Llama-3B and Llama-3B), confirming that cross-backnon setting does not impede and can even amplify gains**.
>
>
> > Q1: Can you describe the sorting process (Curriculum Sort).
>
> A: Thank you for your question regarding CRM. We delineate the pseudo-code of CRM in Appendix B.2 and make a description of the CRM workflow as follows:
>
> - CRM consists of two core components: curriculum learning and peer review. **Specifically, at the beginning of each epoch, we apply curriculum learning to arrange a global easy-to-hard ordering over all training samples, thereby constructing a well-defined learning trajectory. This curriculum serves as a prior that makes RM fit clean and correct data in the early stage to improve generalization.** On the other hand, peer review facilitates collaboration between RMs by evaluating each other’s data selections and filtering out potential noisy preferences at training time.
> - Specifically, the workflow of CRM begins with two reward models $r_{\phi}$ and $r_{\psi}$, training dataset $\mathcal{D}$.
> - Curriculum learning. At each epoch, we rearrange the $\mathcal{D}$ to obtain $\mathcal{D}^{'}$. **Preference pairs are ranked in descending order according to the reward margin (i.e., $r_{\phi}\left(y_w \right)-r_{\phi}\left(y_l \right)$ ), where diminished margin signifies increased difficulty in differentiating response pairs. This allows preference learning into a progression from fundamental patterns to more intricate ones**.
> - Peer review. For each batch $\mathcal{B}$ sampled from the curriculum $\mathcal{D}^{'}$, the two RMs mutually evaluate and select high-quality preferences to update each other:
> $r_{\phi}$ evaluates $\mathcal{B}$ using its BT Loss, and selects the $\lambda_t |\mathcal{B}| $ low-loss pairs to update $r_{\psi}$.
> Here, $\lambda_t$ denotes the selection ratio, defaulting to $1 - \eta$, where $\eta$ is a prior estimator for noise level.
>
> - Similarly, $r_{\psi}$ evaluates $\mathcal{B}$ and selects low-loss pairs for updating $r_{\phi}$. This batch-level refinement facilitates each RM to benefit from the peer’s review, effectively suppressing the noisy preferences.

---

> ### Author Response · Authors · 2025-11-28
> **Kind reminder: updated manuscript**
>
> Thank you again for your time and effort in reviewing our manuscript.
>
> **We have updated the manuscript**, with all changes highlighted in **blue** for the text and captions of revised tables and figures.
>
> If you have any further comments or suggestions, please feel free to share. We are happy to discuss them.
>
> Wishing you a pleasant Thanksgiving!
>
> Sincerely,
> Authors of 5067

---

### Official Review · Reviewer_wBDe · 2025-11-03

**Soundness:** 2
**Presentation:** 2
**Contribution:** 1
**Rating:** 2
**Confidence:** 4

**Summary:**

This paper studies how noisy preference data in RLHF harms reward model generalization and proposes Collaborative Reward Modeling (CRM), where two reward models perform peer review with curriculum learning to filter noisy pairs. Experiments report higher preference accuracy and improved win rates over standard training and robust preference learning baselines, under intentionally added label-flip noise.

**Strengths:**

1. The selected problem setting, i.e., noisy preference data in RLHF is interesting, and is known to impact downstream alignment performance. This paper gives a demonstration through the definition of robust preference pairs, incorrect preference pairs and ambiguous preference pairs, though the the provided results in Figure 2 and 3 may be inaccurate because of "self-loss".
2. The proposed CRM framework is very simple and easy to implement: two reward models select low-loss pairs for each other and follow an easy-to-hard curriculum learning method, which though inevitably brings additional training complexity and costs into the pipeline.

**Weaknesses:**

1. The proposed CRM is limited in theory, and its designs including using self-loss, data filtering strategy and etc. are mostly heuristic, which indicates it may not be effective or extendable in other settings. For example, why choose two models evolve instead of 3 or more?
2. The empirical settings may have fetal flaw that the noise in the data is manually added by symmetric label flipping, which cannot reflect the complex situation in alignment of real world. The used "self-loss" for distinguishing the noised preference pairs may not effective when applied for realistic settings. The effectiveness and performance gains may come from the chosen label flipping. In another way, the self-loss for identifying noisy data is not fair or practical. I would like to know the ratio of noise data in the original preference dataset/benchmark often labeled by human or superior LLMs, and whether how many noisy pairs the self-loss could identity in this situation. In addition, the authors mention that real datasets already include noise, the evaluation does not convincingly separate gains from intentional corruption versus truly realistic noise.
3. The proposed method depends on a prior noise-rate estimate \(\eta\) to set the selection ratio, but this paper gives limited practical guidance on how to estimate this quantity in real deployment.
4. The proposed method treats ambiguous and non-robust pairs as noise to be suppressed, which may introduce additional bias, since ambiguous pairs may not be ambiguous if handled by more powerful LLM or human. Also, the provided cases are very few and limited. Again, it would be better to see the statistics in original dataset.
5. The evaluation is constrained to Llama-3-3B backbones and a single alignment pipeline per setting, so it is unclear whether CRM still brings clear gains at larger model sizes. To my understanding, the reward model should not become the computation burden especially in offline RL settings (e.g., DPO). Also, it will be interesting to see the two models are from different sizes and different model families. I also want to know whether it can be applied to LLM-as-a-Judge setting. It is also suggested to use reasoning models for alignment evaluation instead of relatively outdated GPT-4.
6. To understand the method in depth, It would be useful to see diagnostics or qualitative examples of pairs that are frequently discarded, to understand any systematic bias introduced by the filtering.

**Questions:**

Please see the weakness.

---

> ### Author Response · Authors · 2025-11-18
> **Response to reviewer wBDe (Part 1)**
>
> We are deeply grateful to the reviewer for dedicating valuable time and effort to reviewing our paper.
>
> > Q1: The proposed CRM is limited in theory, and its designs, including using self-loss, data filtering strategy and etc. are mostly heuristic, which indicates it may not be effective or extendable in other settings. For example, why choose two models to evolve instead of 3 or more?
>
> A: Thanks for your thoughtful comments. We will address the design rationale, particularly the use of two models, by first establishing the principles our work is built.
>
> Our approach is motivated by the "memory effect" observed in deep neural networks, which indicates that models first memorize clean, simple patterns before fitting to more difficult or noisy data [1][2]. **Leveraging this property, loss metrics have been widely adopted as a proxy for data cleanliness and an effective indicator for filtering noise [3].** Furthermore, we note that recent studies [4][5] have successfully employed loss functions or gradients to evaluate the quality of preference data.
>
> Built upon these insights, we propose collaborative reward modeling. Specifically, CRM maintains two RMs that refine each other through two key components: Peer Review and Curriculum Learning.
> **Peer review facilitates collaboration between RMs by evaluating each other's data selections and filtering out potential noisy preferences at training time.
> Curriculum learning establishes a well-defined trajectory to synchronize the capabilities of two RMs, further promoting the utility of peer review.**
> These two components work in concert: Peer review provides reliable signals for identifying noisy preferences; curriculum learning synchronizes the capabilities of two models, preventing excessive disparities between two RMs.
>
> To validate the design rationality, we conduct a study by comparing three variants:
>  - w/o Peer Review, which removes the peer review component entirely, leaving only a single reward model trained with curriculum guidance but without noisy data filtering.
>  - w/o Curriculum Learning, the epoch-level curriculum learning module is disabled in CRM, thereby retaining only batch-level peer review during training.
>  - w/ Self Review, uses a single reward model to identify noisy data by itself, and this configuration allows examination of the effects caused by cumulative error and confirmation bias.
>
> |    Noise-level   | CRM |  w/o Peer Review  |  w/o Curriculum Learning | w/ Self Review |
> | -----------      | ---------    | ---------     | --------- | --------- |
> | 0% Flipped       | **77.63**        | 73.72         | 75.06     | 71.24      |
> | 20% Flipped      | **76.19**       | 72.62        | 74.11    | 70.39     |
> | 40% Flipped      | **74.36**       | 71.50         | 72.25   | 67.56     |
>
>
> **By comparing the w/o Curriculum Learning baseline with our CRM approach, we find that CRM leads to more stable behavior in the early stages of training**. The model first captures clean and reliable human preference patterns, which reduces its exposure to noisy annotations. This staged learning process gradually improves generalization and robustness, allowing the model to better identify and discount noisy data later on. **As a result, CRM helps avoid the training instabilities that often arise when noise influences the optimization dynamics throughout the entire process**.
>
>
> [1] Zhang C, Bengio S, Hardt M, et al. Understanding deep learning requires rethinking generalization. ICLR 2017.
> [2] A rpit D, Jastrzębski S, Ballas N, et al. A closer look at memorization in deep networks. ICML 2017.
> [3] Song H, et al. Learning from noisy labels with deep neural networks: A survey. IEEE TNNLS 2022.
> [4] Chowdhury S R, Kini A, Natarajan N. Provably robust DPO: Aligning language models with noisy feedback. ICML 2024.
> [5] Zhang Z, Wang Q, Ye S, et al. Towards Understanding Valuable Preference Data for Large Language Model Alignment. Arxix 2025.

---

> ### Author Response · Authors · 2025-11-18
> **Response to reviewer wBDe (Part 2)**
>
> > Q2: The empirical settings may have a fatal flaw that the noise in the data is manually added by symmetric label flipping, which cannot reflect the complex situation in alignment of the real world. The used "self-loss" for distinguishing the noised preference pairs may not effective when applied for realistic settings. The effectiveness and performance gains may come from the chosen label flipping. In another way, the self-loss for identifying noisy data is not fair or practical. I would like to know the ratio of noise data in the original preference dataset/benchmark often labeled by human or superior LLMs, and whether how many noisy pairs the self-loss could identity in this situation. In addition, the authors mention that real datasets already include noise, the evaluation does not convincingly separate gains from intentional corruption versus truly realistic noise.
>
> A: Thanks for your thoughtful question. In this paper, we focus on the label noise within the preference dataset, which refers to the annotation label that is inconsistent with the ground truth. **Label noise is prevalent in many real-world scenarios and originates from various stages such as data annotation, collection, and processing [1]**. It can also be adversarially manipulated through label-flipping attacks [2], and many works adopt this setup to synthesize noise. Here we report the estimated noise ratios in common preference datasets from [3].
>
> | Dataset      | MT-Bench  | TL;DR | CBArena | SHP |  WebGPT |
> | ----------- | ---------    | ---------     | --------- | --------- |  --------- |
> | Noise rate | 15.0-37.0     | 21.3-27.0         | 22.0-36.0    | 35.5-41.9 |  34.8 |
>
>
> Following [4][5], we employ random flipping to intensify the label noise potentially introduced from different stages of preference construction.
> **In this setting, the dataset contains both the original, inherent noise and the synthetically added noise**.
>
> As shown in Table 1 of our paper, our method effectively improves the generalization of the reward model across various noise levels. **Notably, even without synthetic noise, CRM achieves superior out-of-distribution (OOD) performance on high-quality datasets such as Skywork-Reward and Ultrafeedback.**
>
> Furthermore, we supply the experiment results below on the latest HelpSteer3 without synthetic noise, which is labeled by specialists and rigorously filtered to ensure quality. As shown in the table below, robust preference optimization methods outperform the standard RM, which indicates that the presence of label noise caused by cognitive biases from specialists is inevitable. In addition, **it demonstrates that CRM is effective for reward modeling when dealing with label noise.**.
>
> |             | RewardBench  | RMB-Harmless | RMB-Helpful      | RM-Bench      |
> | ----------- | ---------    | ---------     | --------- | --------- |
> | Standard RM | 68.57        | 63.92         | 70.75     | 56.22     |
> | cDPO-RM     | 70.26        | 65.28         | 69.55     | 57.87     |
> | rDPO-RM     | 70.24        | **67.00**     | 70.05     | 58.10     |
> | ROPO-RM     | 71.32        | 64.17         | 71.33     | 58.57     |
> | CRM         | **74.60**    | 64.79     | **72.61** | **59.32** |
>
>
>
> [1] Song H, et al. Learning from noisy labels with deep neural networks: A survey. IEEE TNNLS 2022.
> [2]Xiao H, et al. Adversarial label flips attack on support vector machines. ECAI 2012.
> [3] Gao Y, Alon D, Metzler D. Impact of Preference Noise on the Alignment Performance of Generative Language Models. COLM 2024.
> [4] Liang X, Chen C, Qiu S, et al. ROPO: Robust Preference Optimization for Large Language Models. ICML 2025.
> [5] Wu  J, et al. Towards Robust Alignment of Language Models: Distributionally Robustifying Direct Preference Optimization. ICLR 2025.
>
> > Q3: The proposed method depends on a prior noise-rate estimate (\eta) to set the selection ratio, but this paper gives limited practical guidance on how to estimate this quantity in real deployment.
>
> A: In our paper, the selection ratio defaults to $1 − \eta$, where $\eta$ is the estimated noise rate of the dataset. For experiments with varying noise levels, $\eta$ is set to the corresponding noise level (20%, 40%). For open-set datasets without label flipping, we set $\eta$ is 10%.
>
> In real deployment, an in-distribution subset（5%） is randomly drawn and subjected to secondary annotation by a superior LLM or human experts. The resulting labels are used to calculate the consistency rate, thereby yielding the estimated noise rate for the whole dataset.

---

> ### Author Response · Authors · 2025-11-18
> **Response to reviewer wBDe (Part 3)**
>
> > Q4: The proposed method treats ambiguous and non-robust pairs as noise to be suppressed, which may introduce additional bias, since ambiguous pairs may not be ambiguous if handled by more powerful LLM or human. Also, the provided cases are very few and limited. Again, it would be better to see the statistics in the original dataset.
>
> A: In Sec. 3.1, we analyze the characteristics of preference pairs in HH-RLHF in Fig.2, with the average loss, loss variance, and predictive accuracy as coordinates. Specifically, **robust preferences represent human-aligned data exhibiting low mean loss and variance, while non-robust preferences arise from annotation mistakes by high losses and low accuracy. Additionally, ambiguous preferences are identified by high variance and nearly 50% predictive accuracy.** Manual inspection indicates this category includes a number of pairs that humans find challenging to differentiate.
>
> **We argue that the statistical distribution of robust, ambiguous, and non-robust preference data is dataset-specific, and their characteristics are vulnerable to annotators, evaluation rubrics, and environmental factors**.
> Limited by cost, it is unfeasible to annotate the entire dataset and obtain an accurate statistical distribution.
> To substantiate the motivation of our approach, we have released more sample cases in Appendix B.1 so that they help readers understand noisy preferences.
>
>
> >Q5: The evaluation is constrained to Llama-3-3B backbones and a single alignment pipeline per setting, so it is unclear whether CRM still brings clear gains at larger model sizes. To my understanding, the reward model should not become the computation burden especially in offline RL settings (e.g., DPO). Also, it will be interesting to see the two models are from different sizes and different model families.
>
> A: We thank the reviewer for raising the important point regarding parameter scaling.
> Here, we utilize two different-family backbones, such as Llama3 and Qwen2.5, then implement the proposed collaborative reward modeling and compare their performance in RewardBench.
>
> - Baseline: Standard RM
>
>     | Standard RM    | Llama3-3B | Qwen2.5-3B | Llama3-8B | Qwen2.5-7B |
>     |----------------------|---------|---------|---------|---------|
>     | RewardBench    | 66.53   | 69.79   | 71.64   | 72.96   |
>
> - CRM: Configure two RMs from different families.
>
>     | CRM     | Qwen 3B & Qwen 3B  | Llama 3B & Llama 3B | Llama 3B & Qwen 3B |
>     |----------------------|------------|-------------|-------------------|
>     | RewardBench    | 77.22      | 77.63       | 78.76             |
>
>     | CRM     | Qwen 3B & Qwen 7B | Llama 3B & Qwen 7B | Llama 3B & Llama 8B |
>     |-------------------|------------------|------------------|-------------------|
>     | RewardBench             | 82.38            | 81.98            | 79.24  |
>
>     | CRM     | Qwen 7B & Qwen 7B | Llama 8B &  Llama 8B |
>     |-------------------|------------------|------------------|
>     | RewardBench|   86.17      |     81.23     |
>
> It can be observed that the Qwen 3B + 7B CRM outperforms a single Qwen 7B by +9.42 points, and even small model pairs like Llama 3B + Qwen 3B exceed the performance of larger single models such as Llama 8B.
> **Meanwhile, CRM still brings clear gains at larger model sizes under Qwen 7B & Qwen 7B and Llama 8B & Llama 8B**.
> Surprisingly, the cross-backbone suite (Llama-3B and Qwen-3B) surpasses the homogeneous backbone suite (Llama-3B and Llama-3B), confirming that cross-backbone setting does not impede and can even amplify gains.
>
>
> > Q6: I also want to know whether it can be applied to LLM-as-a-Judge setting. It is also suggested to use reasoning models for alignment evaluation instead of relatively outdated GPT-4.
>
> A: We appreciate the reviewer’s valuable comment regarding the evaluation. We re-evaluated the TL;DR dataset using DeepSeek-R1 as the Judge LLM. The results are presented in the table below.
>
> | Method | Win  | Tie  | Lose |
> |--------|------|------|------|
> | DPO    | 0.54 | 0.29 | 0.17 |
> | cDPO   | 0.38 | 0.31 | 0.31 |
> | rDPO   | 0.52 | 0.25 | 0.23 |
> | ROPO   | 0.53 | 0.38 | 0.19 |
> | CRM    | **0.56** | 0.28 | 0.16 |
>
> **Experiments show that the reasoning model, when used as a judge, significantly reduces the proportion of "ties".** This indicates it effectively distinguishes between assistant responses, demonstrating a superior capacity for quality comparison.
> Crucially, while the judge is more decisive, this does not change the relative win-rate ordering: **CRM remains the competitive performance**.
>
>
> > Q7: To understand the method in depth, it would be useful to see diagnostics or qualitative examples of pairs that are frequently discarded, to understand any systematic bias introduced by the filtering.
>
> A: Thank you for your careful review. Following your suggestion, we have supplemented more qualitative examples of preference pairs in Appendix B.1, which help readers understand noisy preferences.

---

> ### Author Response · Authors · 2025-11-28
> **Kind reminder: updated manuscript**
>
> Thank you again for your time and effort in reviewing our manuscript.
>
> **We have updated the manuscript**, with all changes highlighted in **blue** for the text and captions of revised tables and figures.
>
> If you have any further comments or suggestions, please feel free to share. We are happy to discuss them.
>
> Wishing you a pleasant Thanksgiving!
>
> Sincerely,
> Authors of 5067

---

### Author Response · Authors · 2025-11-28
**General response**

Dear Reviewers and Area Chair,

We are sincerely grateful for the substantial time and meticulous effort you dedicated to reviewing our submission. The insights, probing questions, and constructive criticisms provided are highly valuable and have been instrumental in significantly elevating the quality and rigor of our manuscript. During this period, we are pleased to receive numerous positive remarks, including:

- **Important research question where the risky impact of noisy preferences on LLM alignment**: wBDe, RgjV
- **Clear motivation and in-depth analysis of the characteristics within noisy preferences**: wBDe, RgjV, 2LFv, 5g3X.
- **Novel, appealing, and well-grounded framework design that targets eliminating the noisy preferences in reward modeling**: wBDe, RgjV, 2LFv, 5g3X.
- **Comprehensive and solid experimental results across diverse training sets and evaluation benchmarks**: wBDe, RgjV, 2LFv, 5g3X.
- **Well-written presentation and informative figures in our manuscript**:2LFv, 5g3X.
- **Strong empirical performance of the proposed method and experimental details are provided**:5g3X, RgjV.
- **Proposed methods that are practical and easy to implement**:wBDe, 5g3X.
- **Broad generalization beyond in-domain, including OOD evaluations**:2LFv, 5g3X.

More importantly, inspired by the reviewers' valuable comments and suggestions, our manuscript has been continually improved regarding the unclear parts or experiments about some specific points. We carefully followed the reviewers' suggestions and supplemented the main additions and clarifications:


- **Ablation analysis of different variants:** supplemented in Section 4.3 and Table 16.
- **Additional experiments on diverse backbones and scales:** supplemented in Section 4.3 and Table 14.
- **Additional analysis of the two-RM collaboration mechanism:** supplemented in Section 4.3 and Table 13 and Table 14.
- **Additional description of noise settings:** supplemented in Appendix B.4 and Table 5.
- **Additional case study of noisy preferences:** supplemented in Appendix B.1.
- **Additional description of the pseudo-code:** supplemented in Appendix C.2 and Algorithm 1 and Table 6.
- **Additional experiments on the latest dataset:** supplemented in Table 15.


Thank you once again for your professional guidance and commitment to enhancing the quality of our work.

Best regards,
Authors of Submission 5067

---

### Meta-Review · Area_Chair_TTJr · 2026-01-06

**Summary:**

This paper proposed a Collaborative Reward Modeling which is a data-centric training scheme where two reward models peer review each other by exchanging low-loss samples, paired with an easy-to-hard curriculum that aims to keep both RMs synchronized and reduce confirmation bias. The method is simple to implement and plug-in to many existing training frameworks.

The main weaknesses are that much of the method is too heuristic, the initial evaluation relied heavily on symmetric label-flip corruption that may not reflect real-world RLHF noise, the method depends on a noise-rate estimate and it’s unclear how robust it is to misestimation, and the addition concerns on the inference time. Several reviewers also demanded fairer baselines, deeper diagnostics about which samples are discarded and any induced bias, and clearer algorithmic details for curriculum sorting and the CRM-DPO extension.

**Reviewer Concerns:**

The authors provide a response in addressing the large-scale and broad model family by introducing more results. However, the following remaining concerns are still open.
1. Most of the response to Reviewer wBDe remains open. For example, it's not well justified why the evaluation is not based on three or more models. Also regarding the self-loss, training time, limited quantitive results are provided.
2. The authors conducted experiments on 7B / 8B models, however this does not fully addressed training on the larger models.

**Reviewer Scores:**

Reviewer wBDe is unlikely to change the score who obtain the most negative response. That being said, the chance of other reviewers to change (increase) their score is slim or at least will not affect the decision of this paper.

---

### Decision · Program_Chairs · 2026-01-26

Reject